# WIDER NETWORKS LEARN BETTER FEATURES

## ABSTRACT

Transferability of learned features between tasks can massively reduce the cost of training a neural network on a novel task. We investigate the effect of network width on learned features using activation atlases — a visualization technique that captures features the entire hidden state responds to, as opposed to individual neurons alone. We find that, while individual neurons do not learn interpretable features in wide networks, groups of neurons do. In addition, the hidden state of a wide network contains more information about the inputs than that of a narrow network trained to the same test accuracy. Inspired by this observation, we show that when fine-tuning the last layer of a network on a new task, performance improves significantly as the width of the network is increased, even though test accuracy on the original task is independent of width.

## 1 INTRODUCTION

While the process of feature learning by deep neural networks is still poorly understood, numerous techniques have been developed for visualizing these features in trained networks in an attempt to better understand the learning process (Erhan et al. (2009); Simonyan et al. (2013); Nguyen et al. (2015); Mordvintsev et al. (2015)). This is usually achieved by optimizing over the input space in order to maximize the activation of a single neuron or filter, with appropriate regularization. These techniques are commonly applied to pre-trained state-of-the-art models, yet they can also be used to explore the effects of various architectural choices on the learned representations. In this work, we specifically investigate the effect of the number of neurons on the quality of the learned representations. It has been shown that increasing the width of a neural network will generally not affect performance on a given task or even improve it (Neyshabur et al. (2017); Geiger et al. (2019)), and it is natural to study the effect of this modification on the learned representations.

We use the recently developed activation atlases technique (Carter et al. (2019)) in order to visualize the inputs that activate an entire hidden state, as opposed to a single neuron. In the case of a wide network, we find that the resulting visualizations differ dramatically.

Our main findings are:

- As we increase the number of neurons in a network, the features that an individual neuron responds to appear less like natural images. However, there exist directions in the hidden state space that are activated by natural-looking images.

- Additionally, the hidden states of wide network contain interpretable information about the inputs that appears not to be present in narrow networks trained to the same test accuracy.

- Fine-tuning a linear classifier on the learned features of a wide network on a novel task leads to a substantial improvement in performance, compared to fine-tuning on the features of a narrow network. This is true even when the test accuracy on the original task is the same for the wide and narrow networks. Presumably, this is a result of the additional information encoded in the hidden states of wide networks.

The last observation suggests that networks designed for transfer learning tasks may benefit from increasing the width, even though this change may have little effect on performance on the original task.

## 2 RELATED WORK

The features that activate individual neurons have been the subject of study for a number of years, with early work focusing on MNIST (Erhan et al. (2009)) as well as more complex image datasets (Simonyan et al. (2013)). The visualization procedure is sensitive to the choice of regularization, and numerous approaches have been explored including regularization using adversarial examples (Szegedy et al. (2013)), robustness to transformations (Mordvintsev et al. (2015)) and using generative models to parameterize the images (Nguyen et al. (2016; 2017)). In contrast to previous works, activation atlases (Carter et al. (2019)) visualize features learned by the entire hidden state. Since these do not focus on single neurons, they are particularly well-suited to studying wide networks where individual neurons may not learn meaningful features.

There is also extensive literature on measuring the responses of individual biological neurons in a number of organisms and brain areas in order to better understand the structure and computation performed in in the input pathways of the brain (Gross et al. (1969); Mohler et al. (1973); Theunissen et al. (2000)).

Thee transferability of learned features between tasks is a well-studied phenomenon (Long et al. (2015); Zoph et al. (2018)). Recently, it was also shown that the performance of the widely used MAML algorithm for transfer learning (Finn et al. (2017)) can be matched on many benchmark tasks simply by fine-tuning the last layer on the new task (Raghu et al. (2019)), which is exactly the fine-tuning setting used in the present work. This suggests that the success of MAML is in learning re-usable features as opposed to converging to points in parameter space that are particularly well-suited to re-training on a novel task.

## 3 ACTIVATION ATLASES

We briefly summarize the visualization technique introduced in Carter et al. (2019). Given a set of inputs $\{x_1, ..., x_k\}$, a network with activations $f^l(x)$ at layer $l$ and input $x$ and a grid size $g$, the technique involves the following steps.

1. Collecting a set of activations $\{f^l(x_1), ..., f^l(x_k)\}$ [1].
2. Reducing the dimensionality of the activations to 2 using UMAP (McInnes et al. (2018)).
3. Defining a 2D grid of size $g \times g$, and averaging over the activation vectors that fall in a given grid cell after dimensionality reduction. The averaged activations are then whitened, giving a set of vectors $\{\widehat{f}^l_1, ..., \widehat{f}^l_{g^2}\}$.
4. For each averaged activation, solving the optimization problem

$$x_i^* = \arg\min_x \left[ g(f^l(x), \widehat{f}^l_i) \right] ,$$

   where $g(x, y)$ is a distance function. Following Carter et al. (2019), we use $g(x, y) = (x \cdot y) \times \max(0.1, \angle(x, y))^4$. The optimization problem is solved iteratively using ADAM (Kingma & Ba (2014)). At every iteration, a series of transformations are applied to $x$ that one expects the network map to be invariant under (such as jitters, rescalings and rotations). This provides a form of implicit regularization during training that improves the visualization results.
5. The resulting images $\{x_1^*, ..., x_{g^2}^*\}$ are then plotted in their respective locations on the grid, giving the activation atlas for layer $l$.

Additional details are provided in Appendix B.1.

## 4 FEATURE VISUALIZATION AND NETWORK WIDTH

In order to visualize the features learned by networks of different widths, we train convolutional networks with different numbers of neurons in the penultimate fully-connected layer, which we

---

[1] When applied to convolutional layers, a spatial location is also chosen at random for each input.

denote by $n$. We then generate activation atlases at this layer, as reviewed in Section 3. The generated images in the atlas are ones which activate the hidden state in a given direction (corresponding to the averaged activations in the given grid cell). As such, they provide an indication of the types of inputs that the network responds to, and the type of structure in the input that it is sensitive to.

We additionally visualize the features that individual neurons respond to by applying the same optimization procedure where the averaged activations $\widehat{f_i^l}$ are replaced by uniformly sampled one-hot vectors. As we will see, there are cases where the hidden state as a whole is activated by interpretable images even though individual neurons are not. Additional experimental details are presented in Appendix B.2.

## 4.1 MNIST

The results on the MNIST dataset for a narrow network with $n = 20$ and for a wide network with $n = 2048$ are shown in Figure 1. Both networks reach $99.0\%$ test accuracy, and as can be seen in the UMAP output the classes cluster well in both cases. However, in the hidden state of the wide network we see additional fine structure that is not present in the narrow network, which can be observed in the atlas images. In the case of the digit 4 for instance, there is a sub-cluster of activations (corresponding to a direction in the hidden state space) that are sensitive to "closed" 4s that contain a triangle. Similarly, there are sub-clusters for 2s with a closed loop and 7s with a crossing line. None of this structure seems to be captured by the hidden state of the narrow network.

Individual neurons in the narrow network respond to images that resemble digits more than those of the wide network. Based on works such as Frankle & Carbin (2018), one may expect that a small subset of neurons in the wide network should still respond to natural images in the same way as the neurons in the narrow network, but this does not appear to be the case. Rather, it appears that the hidden state learns representations that are distributed across all neurons, and as the number of neurons increases the relative amount of information encoded by any individual neuron is reduced.

One may wonder whether these results obtained from the MNIST dataset, which is close to being linearly separable in input space, are in fact trivial. To check this, we generated atlases by using the original images as the activations. The results, presented in Appendix A.2, show that the images are not as clearly interpretable as those generated from the atlases on the trained networks (and as expected, the clustering is not as good).

## 4.2 TRANSLATED MNIST

The difference between the information in the hidden states of narrow and wide networks is easier to discern in datasets with additional fine grained structure compared to MNIST. We construct such a dataset, which we call *translated MNIST*, by selecting a subset of training images from MNIST and horizontally translating each image cyclically in all possible 28 ways. The size of the training set remains the same, and the test set is left unchanged. We then generated the atlases for this dataset using the same network architecture and training procedure as in the previous subsection. The final test accuracies were $96.0\%$ and $96.8\%$ for widths $n = 20$ and $n = 2048$, respectively.

As can be seen in Figure 2, the additional structure induced by the translations is clearly present in the hidden state of the wide network in the form of the annular structures in the UMAP output. The location of activation vectors in an annulus for a given class are seen to correspond to the degree of translation of the image. In contrast, the hidden state of the narrow network does not clearly encode the degree of translation. Since the translation information was present in the inputs, one may wonder whether this gap between the wide and narrow network is simply a result of a bottleneck in the narrow network. As we show in Section 5 however, the features learned by the wide network are more useful than either the features of the narrow network or the inputs themselves when fine-tuning a linear classifier on a new task.

## 4.3 CIFAR-10

Similar phenomena to those described above are also observed on the CIFAR-10 dataset. We again train convolutional networks of the same architecture, this time varying not only the width of the final layer but of all intermediate layers as well by scaling the number of filters in the convolutional

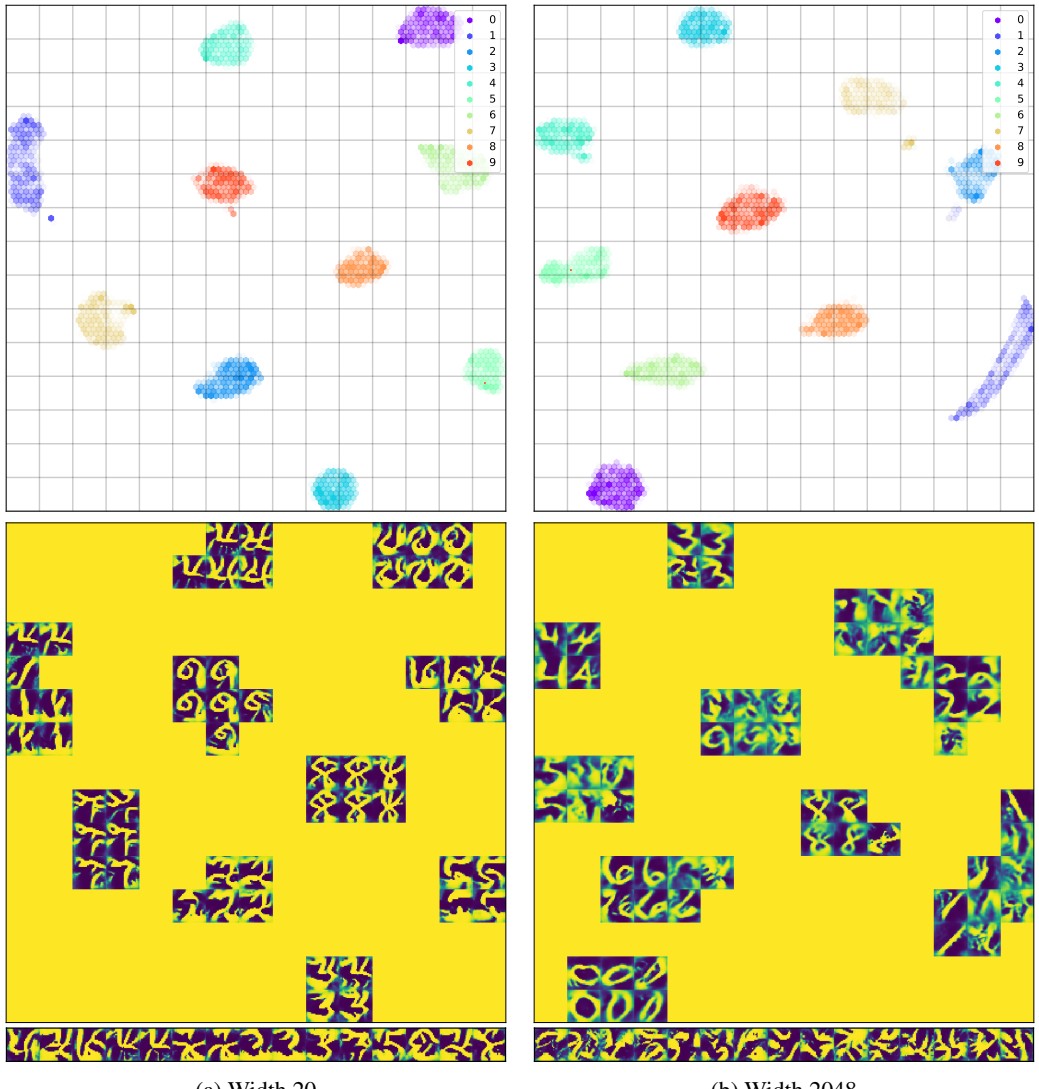

(a) Width 20                                (b) Width 2048

Figure 1: Activation atlases for the penultimate fully-connected layer of narrow and wide convolutional networks trained on MNIST. *Top:* 2D histograms of the output of UMAP applied to the activation vectors, with color corresponding to class label. While both networks cluster the inputs according to class as expected, there are additional sub-clusters in wide network activations. *Middle:* Activation atlases, indicating the representations learned by the networks. The sub-clusters observed in the top plot for wide networks can be seen to correspond to digits with distinct features. *Bottom:* The features that randomly selected individual neurons respond to. For the narrow network, the individual neuron and the activation atlas features appear to be equally interpretable, while for the wide network the individual neuron features are not interpretable.

layers linearly with $n$. The final test accuracies for the networks of width $n = 64$ and $n = 2048$ were 83% and 82% respectively.

The results are presented in Figure 3, where there again appears to be more information in the hidden state of the wide network than the narrow one. Similar to the fine structure observed in the atlases for MNIST and translated MNIST (which enabled one to distinguish the degree of translation or style of digit), here we can distinguish between automobiles of different color in the lower left section of the atlas (and not in the case of the narrow network). On the right section of the atlas, we see that left- and right-facing horses become separated. As before, we also observe that individual neurons learn less interpretable features in the wide network.

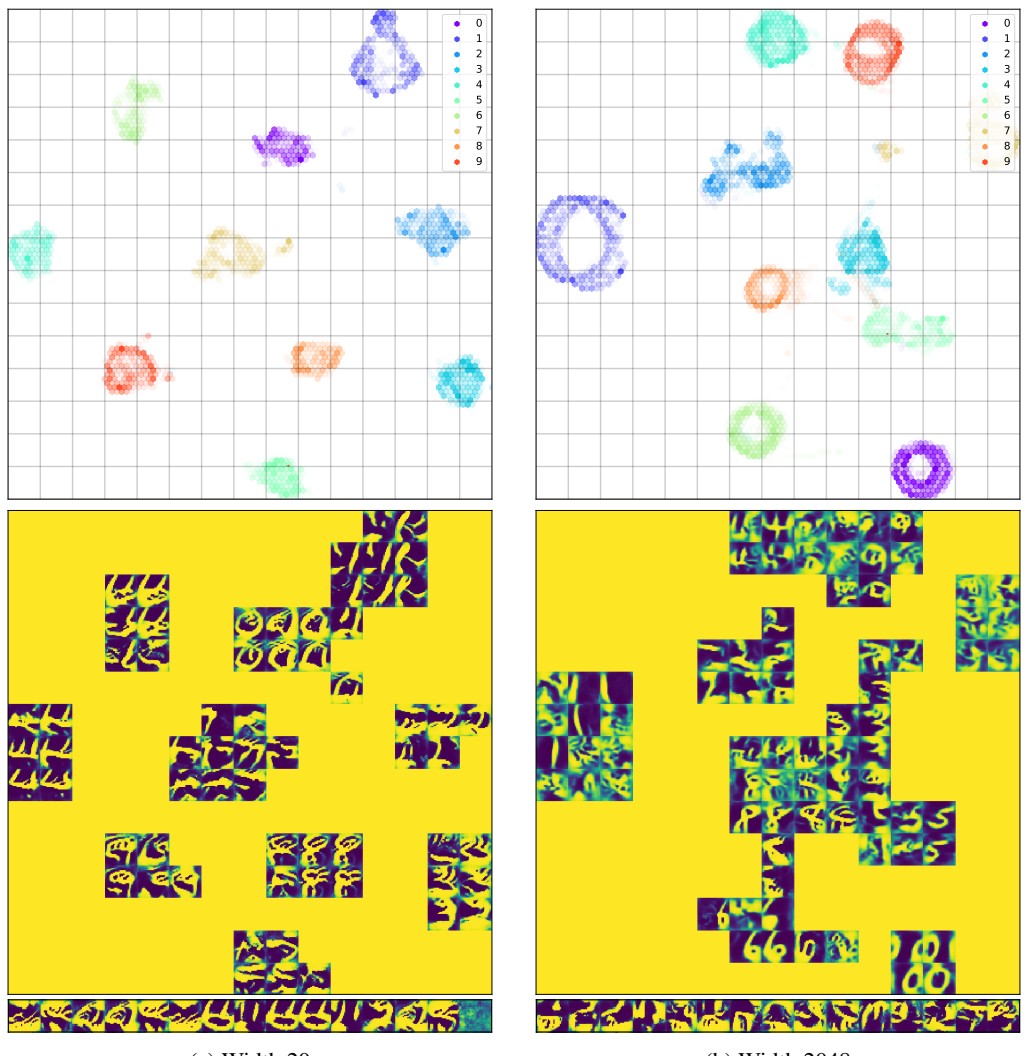

(a) Width 20                             (b) Width 2048

Figure 2: Activation atlases for the penultimate fully-connected layer of convolutional networks of different width trained on translated MNIST. *Top:* 2D histograms of the output of UMAP applied to the activation vectors, with color corresponding to class label. Compared to MNIST (Figure 1), the additional structure due to the translations is clearly visible in the wide network but not in the narrow one. *Middle:* Activation atlases. For the wide network, the location of the activations in an annulus for a given class is indicative of the degree of translation of the image. *Bottom:* The features that randomly selected individual neurons respond to. As in the case of MNIST, these are more interpretable for the narrow network.

## 5   USING LEARNED REPRESENTATIONS FOR NOVEL TASKS

The additional structure present in the hidden states of wide networks, seen in Figures 1, 2, is not necessary to perform well on the original classification task. This is attested by the fact that the performance of the two networks on the original task was nearly identical. However, the presence of the additional structure suggests that wide networks might perform better on novel tasks that rely on the additional information. We test this conjecture by training a network of depth $L$ on one task, obtaining a trained network

$$f_{\text{orig.}}(x) = W_{\text{orig.}}^L f_{\text{orig.}}^{L-1}(x) \,,$$

and then fine-tuning a linear classifier that takes as input the features of the original network,

$$f_{\text{tuned}}(x) = W_{\text{tuned}}^L f_{\text{orig.}}^{L-1}(x) \,.$$

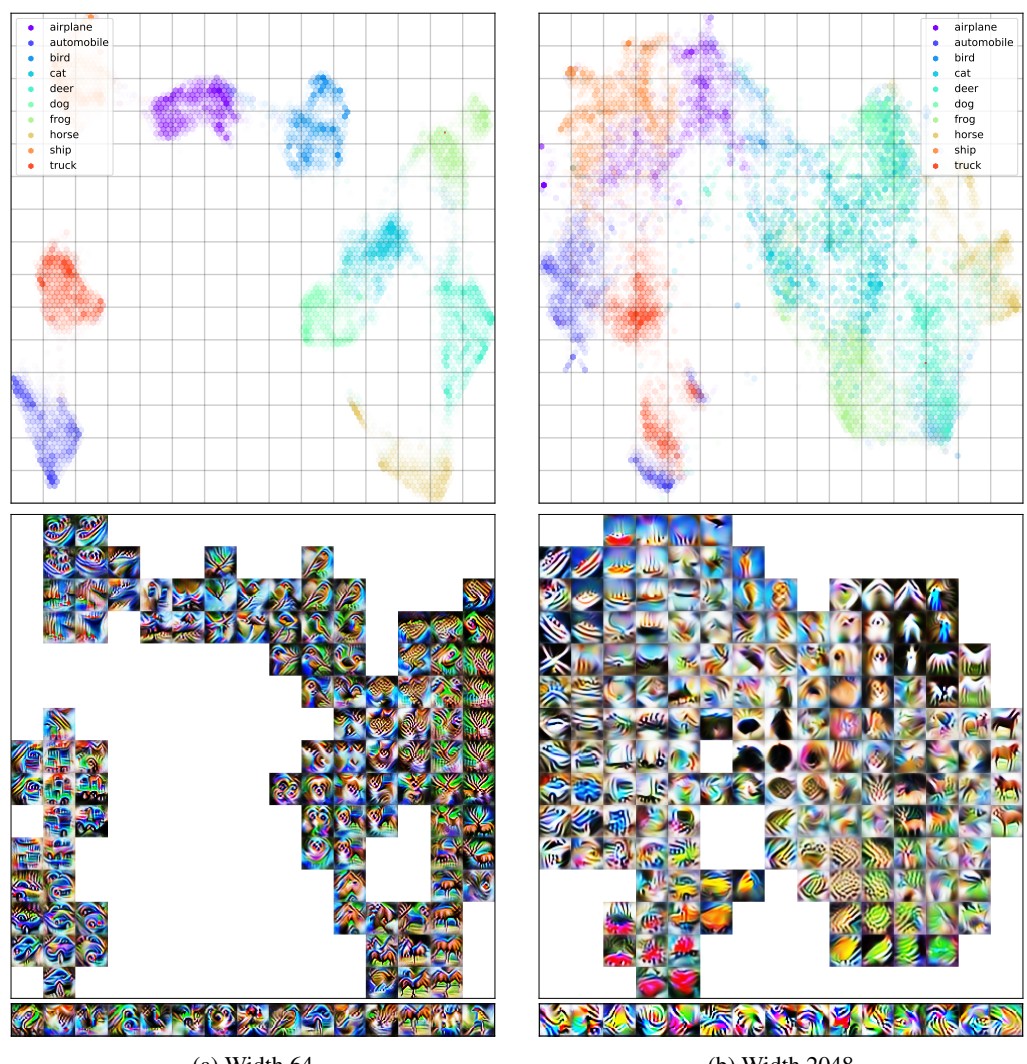

(a) Width 64          (b) Width 2048

Figure 3: Activation atlases for the penultimate fully-connected layer of convolutional networks of different width trained on CIFAR-10. *Top:* 2D histograms of the output of UMAP applied to the activation vectors, with color corresponding to class label. *Middle:* Activation atlases. For the wide network, there is a direction of large variance in the activation space corresponding to automobile color in the bottom left, and to left/right orientation of horses on the right. *Bottom:* The features that randomly selected individual neurons respond to. As before, these are more interpretable for the narrow network.

In training $f_{\text{tuned}}(x)$, all the weights in $f_{\text{orig.}}^{L-1}$ are frozen. The output dimension can also change in the fine-tuned network. Additional experimental details are presented in Appendix B.3.

## 5.1 TRANSLATED MNIST — DIGIT AND TRANSLATION CLASSIFICATION

We first train fully connected networks with varying width and depth to predict the degree of translation of an image in the translated MNIST dataset, described in Section 4.2. We refer to this as the *original task*. The details of the network architecture and training procedure are specified in Appendix B.2. As we reduce the width, we also increase the depth so that the total number of model parameters is approximately constant. The widest network in our experiment has 1 hidden layer with 2000 neurons, and the narrowest has 20 hidden layers with width 250. We observe that test accuracy after training on the original task (translation classification) is essentially independent of width. These results are shown in the left panel of Figure 4.

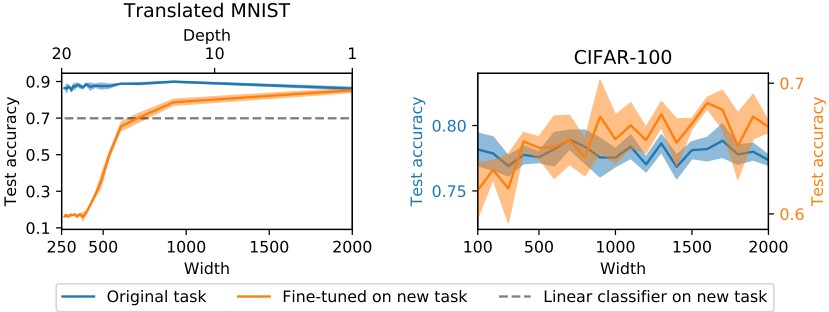

Figure 4: Wide networks learn features that are useful on novel tasks. We train networks with different numbers of features at the penultimate layer on one task, then fine-tune the last layer on a novel task. Performance on the original task is essentially independent of width, but performance of the fine-tuned classifier improves dramatically with increased width. The results are the mean and standard deviation over 5 initializations. *Left:* Translated MNIST (described in Section 4.2). The original task is shift classification, and the new task is digit classification. The networks are fully connected and the total number of parameters is held approximately constant (by making the narrow networks deeper). *Right:* 3 coarse class CIFAR-100, convolutional networks. The original task is image classification by 3 coarse classes (superclasses) of CIFAR-100. The new task is classification by the fine classes of the same images. A linear classifier trained directly on the new task achieves test accuracy $37.5\%$.

Next, we fine-tune the last layer of the network to classify the digits (as in standard training on MNIST). The number of classes is reduced from 28 to 10. We refer to this as the *new task*. We find that performance improves dramatically with increased width, while both the test accuracy on the original task and the number of parameters are approximately constant (see the orange curve in the left panel of Figure 4).

Some of this effect may be due to the fact that the narrower networks have a bottleneck. Indeed, we see that performance on the new task increases rapidly until the width reaches the input dimension $w_* = 784$, and beyond that point the improvements are more gradual. The performance achieved at width $w_*$ is also the performance of a linear classifier trained on the new task (the dashed line in the left panel of Figure 4). The fact that wider networks achieve better performance on this task suggests that they are learning useful features.[2]

## 5.2 CIFAR-100: Coarse and Fine Label Classification

We perform a similar experiment on the CIFAR-100 dataset. Each image in the CIFAR-100 dataset has both a fine and a coarse label. There are 20 coarse labels and 100 fine sub-labels, where each coarse label is associated with 5 fine sub-labels (the aquatic mammal coarse class, for example, contains images in the fine classes beaver, dolphin, otter, seal, and whale). We use images in the first 3 coarse classes only, giving a training set of $7,500$ images, and a test set of $1,500$ images. We train convolutional networks with wide or narrow penultimate hidden layers to classify images according to the coarse label — the original task in the right panel of Figure 4. We then freeze these networks, and fine-tune the last layer to classify images according to the fine labels — this is the new task in the right panel of Figure 4. We record the best test accuracy achieved during training by the fine-tuned classifier.

As in the case of fully-connected networks on the translated MNIST task, we find that the test accuracy on the original task essentially does not depend on the number of neurons for the range of widths considered.[3] We again observe that the test accuracy of the fine-tuned classifier increases

---

[2]We expect that digit classification on the translated MNIST dataset is harder than the original MNIST task. It is therefore not surprising that the linear classifier does worse on this task than on original MNIST, where $90\%$ test accuracy is achievable.

[3]It is possible that by tuning hyperparameters such as learning rate one can improve the performance of the wider networks on the original task (Geiger et al. (2019)). We have not explored this possibility.

with width. A linear classifier trained directly on the inputs achieves a test accuracy of only $37.5\%$ on the new task.

## 6 DISCUSSION

Visualization of the inputs that activate the entire hidden state of a network, as opposed to individual neurons, suggests that wide networks contain more information in their hidden state than narrow networks, even if the two networks are trained to the same test accuracy. This is true even when controlling for the total number of parameters. This observation, a result of generating activation atlases (Carter et al. (2019)) for networks of varying widths, suggests that wide networks may perform better than narrow ones when trained on novel tasks. We indeed observe this effect when fine-tuning the last layer of a network on new labels. Considering our findings in the light of the recent results of Raghu et al. (2019), which suggest that feature re-use is the dominant factor explaining the success of MAML (Finn et al. (2017)), it will be interesting to explore the effect of network width on the performance of this and other transfer learning algorithms.

We find that the activation vectors of wide networks are not sparse, and the visualization of features learned by individual neurons indicate that there isn't a small subset of neurons that respond to natural images. These results suggest that information is stored in a distributed fashion in the hidden state of a network. This may be surprising given recent work indicating that network performance is only slightly degraded after zeroing out most of the weights (Frankle & Carbin (2018)).

Our empirical observation that there is information in the hidden state that appears to not be useful for classification (such as the degree of translation in translated MNIST, see the middle right panel in Figure 2) also appears to be at odds with the view that neural networks perform compression in later stages of training. According to this picture, training reduces the mutual information between the hidden state and the inputs while maintaining the mutual information between the hidden state and the labels (Schwartz-Ziv & Tishby (2017)). Conversely, the picture that emerges from our experiments is that when the network has the capacity to do so, it can transform the data manifolds in order to enable classification, without destroying additional structure in those manifolds that is not required for classification. Furthermore, it appears this additional capacity is more closely tied to larger width than to larger depth. It may still be the case that if a bottleneck is induced in the network (say by introducing an intermediate narrow layer) such information will be lost.

Finally, we would like to discuss our observations in the context of very wide neural networks, which have been studied for example in Neal (1996); Daniely et al. (2016); Lee et al. (2018); de G. Matthews et al. (2018). In the infinite width limit, when the network is properly parameterized, activation vectors change by a negligible amount compared with their initial values (Du et al. (2018b); Li & Liang (2018); Zou et al. (2018)). This is sometimes referred to as *lazy training* (Chizat et al. (2018)). In this regime, we expect that activation atlases will not produce interpretable results, because the activations are almost the same as those at initialization. This raises the question of whether very wide networks learn interpretable features at all, and if so, how are these features encoded in the hidden state. Answering these questions in full is beyond the scope of this work, but we will make a few comments that we hope may shed light on this question.

Consider a network function $f(x) = W^L f^{L-1}(x) \in \mathbb{R}^k$, where $f^{L-1}(x) \in \mathbb{R}^n$ is the activation in the penultimate layer, $W^L \in \mathbb{R}^{k \times n}$ are the weights of the linear classifier at the head of the network, and $k$ is the number of classes. As we take the width $n$ large, and with appropriate parameterization, the learned activations $f^{L-1}(x)$ will remain close to their initial values as measured (for example) by $L_2$ distance. However, note that the classifier only makes use of a $k$-dimensional subspace of the hidden space $\mathbb{R}^n$, spanned by the rows of $W^L$. While the whole activation vector $f^{L-1}(x)$ does not move by much during training, the activation vector projected onto this smaller subspace can move by a significant amount, and it is only this projection that matters for the purpose of classification. Therefore, it is possible that learned, interpretable features are encoded in the hidden state projected to an appropriate subspace, even though the hidden state itself remains very close to its initial value. Correspondingly, it is possible that activation atlases can produce interpretable results when trained on these projected activations. Of course, the classifier subspace itself is updated during training, and so these learned features may in fact be encoded in a larger subspace defined by the history of classifier subspaces seen during training. We leave a more careful treatment of this question to future work.

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

## A ADDITIONAL ACTIVATION ATLASES

### A.1 UNTRAINED MODELS

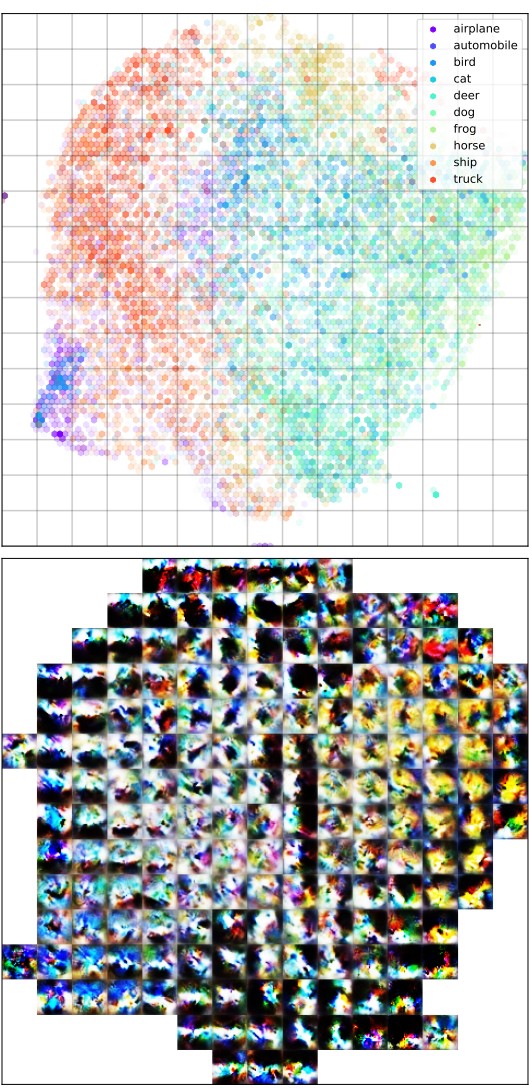

Figure 5: Activation atlas for the penultimate fully-connected layer of an untrained convolutional network. The atlas images suggest that the hidden states that are activated by images of a similar color are adjacent to each other, and this correlates with the class label to an extent.

In Figure 5 we show the activation atlas for an untrained convolutional network. The architecture is described in Appendix B.2, with the width of all layers set to $128$ except the penultimate fully connected layer of width $512$. Unsurprisingly, there is no clustering in the UMAP output and the resulting atlas images are not interpretable. This indicated that obtaining interpretable atlas images is a consequence of learning features from data.

### A.2 ATLAS IN INPUT SPACE

Figure 6 shows the activation atlas generated for raw MNIST inputs. The averaged activations $\widehat{f}_i$ in this case are simply whitened averages over images. While the inputs clearly cluster indicating that the data is nearly linearly separable, a comparison to Figure 1 shows that the clusters are not as well separated as in the case of trained networks, and the atlas images produced are less interpretable. In particular, sub-clusters corresponding to fine-grained features are not visible.

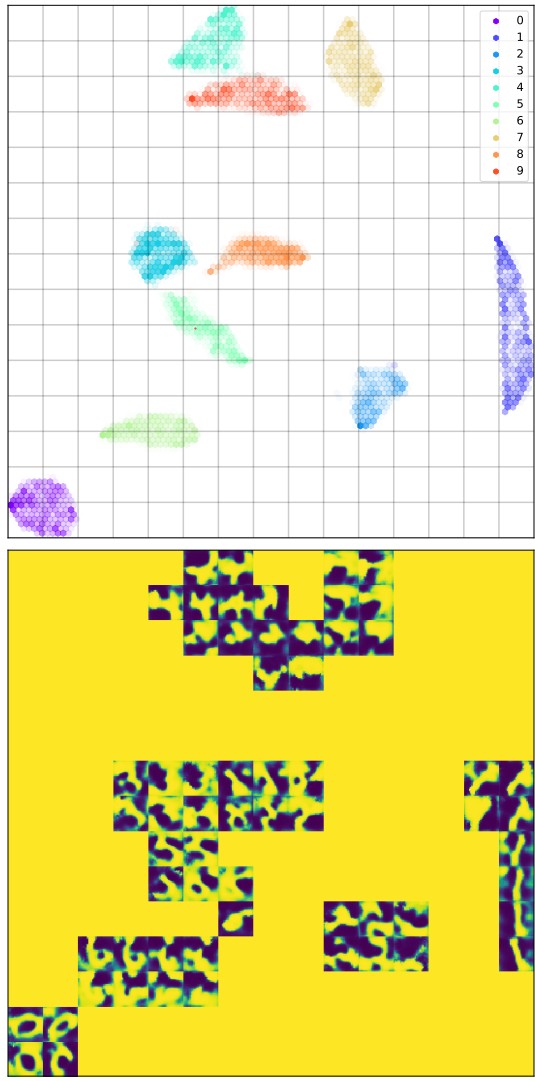

Figure 6: Activation atlas for MNIST digits. The class clusters are not as well separated as those in a trained network, and the atlas images are less interpretable.

## B EXPERIMENTAL DETAILS

### B.1 ACTIVATION ATLASES

The inputs used to generate the atlases were the entire training set for all datasets. The grid size $g$ was set to 15.

The code to generate the activation atlases was based on `https://github.com/tensorflow/lucid`. The regularization applied at every iteration, the UMAP parameters and the optimization hyper-parameters were identical to those in the code. MNIST digits were padded with 0, while CIFAR-10 images were padded with 1.

### B.2 NETWORK ARCHITECTURES AND TRAINING DETAILS

The network architecture consists of two blocks of two convolutional layers each, with filter size 3 followed by a max-pooling layer with pool size 2. These are followed by two fully-connected layers. The networks use ReLU nonlinearities and the weights drawn from $W_{ij}^l \sim \mathcal{N}(0, \frac{2}{n_{\text{in}}})$ for all

but the last layer and $W_{ij}^L \sim \mathcal{N}(0, \frac{1}{n_{\text{in}}})$, while the biases are initialized at $0$. Here, $n_{\text{in}}$ is the fan-in dimension (suitably defined for the convolutional layers to include the filter size).

All networks were trained using cross-entropy loss with SGD with momentum ($\beta = 0.9$), using batch size 128. The training set size was $50,000$ for MNIST and CIFAR-10, and $28 \times \lfloor 50,000/28 \rfloor$ for translated MNIST (see Section 5.1). The test set consisted of $10,000$ images for all datasets. Networks trained on MNIST and translated MNIST were trained for 10 epochs with a learning rate of $0.01$, while networks trained on CIFAR-10 were trained for 100 epochs with a learning rate of $1/n$, where $n$ is the width of the network. CIFAR-10 data was also augmented with random shifts of up to $10\%$ both horizontally and vertically, rotations of up to $\pi/6$ and horizontal flips.

### B.3  FINE-TUNING TASKS

#### B.3.1  TRANSLATED MNIST

The scan in Figure 4 is of fully-connected networks with 1 to 20 hidden layers and width chosen such that the number of parameters is approximately constant. Denoting by $n^{(L)}, n_0, n_{L+1}, n_{\max}$ the width for the $L$ hidden layer network, input dimension, output dimension and maximal width (set to 2000) respectively, we set $n^{(L)}$ according to

$$n^{(L)} = \left\lfloor \frac{-L - n_0 - n_{L+1} + \sqrt{(L + n_0 + n_{L+1})^2 + 4(L-1)(n_0 + n_{L+1} + 1)n_{\max}}}{2(L-1)} \right\rfloor .$$

The learning rate used for all widths was 0.01. This was obtained from a scan across $10^x$ for 5 linearly spaced values of $x$ between $-1$ and $-3$, since it led to the best performance across different depths.

#### B.3.2  CIFAR-100

The training set and test set consisted of images in thee coarse classes aquatic mammals, fish and flowers. Networks were trained on coarse label classification for 100 epochs. The learning rate used for all widths was 0.0129. This was obtained from a scan across $10^x$ for 10 linearly spaced values of $x$ between $-1$ and $-3$, since it led to the best performance across different depths on average.

