# OpenReview forum: "Wider Networks Learn Better Features"
_ICLR.cc/2020/Conference — Reject_

### Official Review · AnonReviewer1 · 2019-10-20
**Official Blind Review #1**

**Rating:** 3

**Review:**

This paper considers the effect of network width of the neural network and its ability to capture various intricate features of the data. In particular, the central claim of this paper is what the title claims "Wider networks learn features that are better". They make this claim using the visualization technique called "activation atlasses". They find that wider networks learn features in the hidden neurons that are more "interpretable" in this visualization framework. Additionally, they also notice that fine-tuning a _linear model_ using the learned features for the wider networks provide better accuracy for new (but related) tasks over the shallower counterparts. For most experiments of this paper, "shallow network" refers to a width of 64 and "wide network" refers to a width of 2048. The main datasets used for the experiments are MNIST, CIFAR 10/100 and a "translated" version of MNIST images.


Overall the paper is written well and the ideas and results are communicated crisply. I have a few comments. First, regarding the related work, I think that the reader would be served better if the authors also list the recent works related to effect of network width on convergence and generalization (e.g., [1] and references that cite this). The reason I say this is so that the reader should not (wrongly) interpret that this is the first work that finds "favorable" properties of wider networks (the paper does not make this claim, but it is easy for a reader to interpret it). Second, I find it slightly concerning that a lot of findings have been extrapolated from just one architecture. In particular, I find the experiments in section 5 to be the most informative (and also objective), since it is a single number which is easy to think about. To be clear, I like the visualization experiments and it gives credibility to the claim about interpretability. Given that there are many levers in a neural net (batch norm, architectural choices, hyper-params etc.) one could fiddle with, to make the claim made in the introduction one needs a more extensive set of experiments. I acknowledge that the authors say they haven't explored the possibility of fine-tuning the hyper-params for instance, but I think considering some of these choices is really helpful. This will help _isolate_ the effect of width independent of the architecture choice.

Given the above observations, my current decision of this paper is that it doesn't meet the bar. I find the results promising but the paper is not yet ready.


[1] - https://papers.nips.cc/paper/8076-neural-tangent-kernel-convergence-and-generalization-in-neural-networks.pdf

**Experience Assessment:**

I have published one or two papers in this area.

**Review Assessment: Checking Correctness Of Derivations And Theory:**

N/A

**Review Assessment: Checking Correctness Of Experiments:**

I assessed the sensibility of the experiments.

**Review Assessment: Thoroughness In Paper Reading:**

I read the paper at least twice and used my best judgement in assessing the paper.

---

### Official Review · AnonReviewer3 · 2019-10-23
**Official Blind Review #3**

**Rating:** 1

**Review:**

The authors observed that the wider deep neural networks can learn much rich representative features than shallower deep neural networks while both networks show similar level of the test performance. They show feature visualization about their observations using two different networks n=20/n=2048.

At the first, I feel that the visualizations on figure 1 about two different width are too marginal. Almost they look similar, it's hard to say that significantly show difference.

Also there is no guarantee that the quality of the feature visualization follows linear relationship according to width. Comparison with just two different width is not enough to analyze the situation.

I wonder if human-interpretable features are always better. Machine-interpretable information also do important role, as adversarial attack.

Even the total number of parameter is preserved, the performance will be largely vary according to the network architecture, such as the number of the layers. Then, I have a doubt whether experiments on Sec 5.1 are meaningful not. More, the model can suffer from the gradient vanishing problem when the network has a number of layers. I wonder that the results on figure 4 are caused from this problem.





**Experience Assessment:**

I have published one or two papers in this area.

**Review Assessment: Checking Correctness Of Derivations And Theory:**

I assessed the sensibility of the derivations and theory.

**Review Assessment: Checking Correctness Of Experiments:**

I carefully checked the experiments.

**Review Assessment: Thoroughness In Paper Reading:**

I read the paper thoroughly.

---

### Official Review · AnonReviewer2 · 2019-10-23
**Official Blind Review #2**

**Rating:** 3

**Review:**

This paper investigates wider networks using a recent feature visualization technique named activation atlases. By analyzing what the hidden layers of wider networks respond to, the authors showed that wider networks learn more transferable features. However, I tend to reject this paper since it doesn’t show very compelling evidence through experiments.

1. This paper does not present any novel methods, and so the experiments need to be very solid. But all the datasets and architectures used in this paper are quite simple from the view of deep learning. It is not clear whether the conclusions will still be valid for larger datasets or deeper networks.

2. The most important observation of this paper is to find that wider networks can be easily transferred to a new task. But in Section 5.1, all layers except the last classification layer are fixed when fine-tuning the networks for the second task. It is so obvious that wider networks with fewer previous layers can perform better. In Section 5.2, the authors did not show the network details, and also it is not fair to compare the networks with a linear classifier. The authors should include more competitive baselines.

3. I encourage the authors to show the training/validation curves to testify the data efficiency of the wider networks when training or fine-tuning on a new task.

**Experience Assessment:**

I have read many papers in this area.

**Review Assessment: Checking Correctness Of Derivations And Theory:**

N/A

**Review Assessment: Checking Correctness Of Experiments:**

I carefully checked the experiments.

**Review Assessment: Thoroughness In Paper Reading:**

I read the paper at least twice and used my best judgement in assessing the paper.

---

### Decision · Program_Chairs · 2019-12-19

**Decision:**

Reject

**Comment:**

This paper investigated the effect of network width on learned features using activation atlases. From the current view of deep learning, the novelty of the paper is limited.

As all reviews rejected the paper and the authors gave up rebuttal, I choose to reject the paper.